# Genetic Dissection of Light-Regulated Adventitious Root Induction in *Arabidopsis thaliana* Hypocotyls

**DOI:** 10.3390/ijms23105301

**Published:** 2022-05-10

**Authors:** Yinwei Zeng, Sebastien Schotte, Hoang Khai Trinh, Inge Verstraeten, Jing Li, Ellen Van de Velde, Steffen Vanneste, Danny Geelen

**Affiliations:** 1Department Plants and Crops, Faculty of Bioscience Engineering, Ghent University, Coupure Links 653, 9000 Ghent, Belgium; yinwei.zeng@ugent.be (Y.Z.); sebastien.schotte@gmail.com (S.S.); hoangkhai.trinh@ugent.be (H.K.T.); inge.verstraeten@gmail.com (I.V.); jingjili.li@ugent.be (J.L.); ellen.vandevelde@ugent.be (E.V.d.V.); 2Biotechnology Research and Development Institute, Can Tho University, Can Tho City 900000, Vietnam; 3Department of Plant Biotechnology and Bioinformatics, Faculty of Sciences, Ghent University, Technologiepark 71, 9052 Ghent, Belgium; 4VIB Center for Plant SystemsBiology, VIB, Technologiepark 71, 9052 Ghent, Belgium; 5Lab of Plant Growth Analysis, Ghent University Global Campus, Incheon 21985, Korea

**Keywords:** adventitious root, hypocotyl, photomorphogenesis, light, *Arabidopsis thaliana*

## Abstract

Photomorphogenic responses of etiolated seedlings include the inhibition of hypocotyl elongation and opening of the apical hook. In addition, dark-grown seedlings respond to light by the formation of adventitious roots (AR) on the hypocotyl. How light signaling controls adventitious rooting is less well understood. Hereto, we analyzed adventitious rooting under different light conditions in wild type and photomorphogenesis mutants in *Arabidopsis thaliana*. Etiolation was not essential for AR formation but raised the competence to form AR under white and blue light. The blue light receptors CRY1 and PHOT1/PHOT2 are key elements contributing to the induction of AR formation in response to light. Furthermore, etiolation-controlled competence for AR formation depended on the COP9 signalosome, E3 ubiquitin ligase CONSTITUTIVELY PHOTOMORPHOGENIC (COP1), the COP1 interacting SUPPRESSOR OF PHYA-105 (SPA) kinase family members (SPA1,2 and 3) and Phytochrome-Interacting Factors (PIF). In contrast, ELONGATED HYPOCOTYL5 (HY5), suppressed AR formation. These findings provide a genetic framework that explains the high and low AR competence of *Arabidopsis thaliana* hypocotyls that were treated with dark, and light, respectively. We propose that light-induced auxin signal dissipation generates a transient auxin maximum that explains AR induction by a dark to light switch.

## 1. Introduction

Light, as with many other environmental factors, is of fundamental importance for the autotrophic growth of plants [1]. Therefore, it is not surprising that plants tune their body plan through the correct positioning of aerial organs or to accelerate elongation to outgrow competing plants to optimally capture and use the available light [2,3]. In fact, light triggers a major developmental transition just after germination [4]. Seedlings that germinated in the dark, grow almost exclusively by elongation in their search for light, which in turn signals that they have reached the soil surface. This is referred to as skotomorphogenesis. Once seedlings emerge into the light, plants develop a de-etiolated morphology (photomorphogenesis), including a short hypocotyl and open, expanded and green cotyledons [5], and the development of adventitious roots (AR) [6] The latter is, however, a poorly understood effect that is associated with the induction of photomorphogenesis. In fact, light has been reported to both stimulate and inhibit rooting [7]. While the dark to light switch is a commonly used AR induction assay [8], it was recently reported that continuous dark treatment is sufficient for AR formation [7]. Moreover, different light conditions, in terms of intensity and quality, also differentially impact on AR formation [9,10]. These examples illustrate that light and/or photomorphogenesis pathways exert complex effects on AR formation that remain to be resolved.

The transition from skoto- to photomorphogenesis is regulated by a diverse group of photoreceptors that allow plants to monitor and respond to different wavelengths of ambient light [11]. When grown under light, the photoreceptor PHYA is rapidly degraded, and the effects of PHYB and the cryptochromes begin to dominate [12]. Blue light-independent cryptochromes CRY1 and CRY2 are predominantly nuclear proteins that play major roles in photomorphogenesis [13,14], mediate entrainment of the circadian clock in response to light [15,16] and regulate up to 10–20% gene expression in the *Arabidopsis thaliana* genome [17]. As well as cryptochromes, two phototropins, designated PHOT1 and PHOT2, are blue light receptors in *Arabidopsis*
*thaliana* [18], and both act to mediate the phototropism of hypocotyl in response to high intensities of blue light [19]. Recently, PHOT1 and PHYB were implicated in AR formation by, respectively, modulating auxin transport in response to blue light [9] and inhibiting auxin signaling in the dark [7]. These examples clearly connect light perception with the control of AR formation, but a mechanistic model is lacking.

The E3-ligase CONSTITUTIVELY PHOTOMORPHOGENIC1 (COP1) and SUPPRESSOR OF phyA-105 (SPA) kinases prevent photomorphogenic growth in the dark [20]. In *Arabidopsis*
*thaliana*, the SPAs bind and activate the E3 ubiquitin ligase COP1 [21,22] to target the transcription factor ELONGATED HYPOCOTYL 5 (HY5) for 26S proteasome-mediated degradation [22,23,24]. This activity of COP1 requires its nuclear localization, which depends on the action of the COP9 signalosome (CSN) [25,26,27]. The CSN is an evolutionary conserved eight-subunit protein complex originally identified as a genetic screen for mutants that mimic light-induced photomorphogenic development when grown in the dark [25,28]. CSN controls the RUB/NEDD8 modification of the CULLIN subunit of multiple E3 ligases, thereby regulating not only plant photomorphogenesis [29] but also a set of diverse signaling cascades such as DNA damage [30], auxin signaling [31,32,33,34] and jasmonate signaling [35]. Mutants in different CSN subunits were shown to be defective in auxin-induced AR formation [36]. COP1, HY5 and the CSN complex act downstream of photoreceptors such as PHYA, PHYB, CRY1, CRY2 and PHOT2 [11,37,38,39]. Their role and interplay during adventitious rooting is, however, poorly characterized.

Photomorphogenic development is largely driven by these factors, converging on extensive transcriptional reprogramming, that is mediated by HY5 and PHYTOCHROME-INTERACTING FACTOR (PIF) transcription factors [40,41]. PIF transcription factors are stable in the dark and degraded in the light [41], and they suppress photomorphogenesis [42,43]. PIFs are antagonized by the basic leucine zipper (bZIP) transcription factors HY5 and its homolog HYH [44] that are stable in the light to promote photomorphogenesis [45,46,47,48]. The most prominent phenotype of skotomorphogenic development is hypocotyl elongation and apical hook formation [4]. In the light, HY5 represses hypocotyl elongation by enhancing brassinosteroid signaling [48], mediating ultraviolet-B (UVB) radiation [49] and regulating transcriptional changes [50,51]. The apical hook formation is regulated by PIFs, in parallel with ETHYLENE INSENSITIVE3 (EIN3) and EIN3-LIKE1 (EIL1) transcription factors in response to hormone and light signals [24,52,53,54]. Of these transcription factors, EIN3/EIL1 was reported to suppress adventitious rooting from leaf explants [55], while the roles of HY5 and PIFs in AR formation remained to be evaluated.

The plant hormone auxin is a potent regulator of plant growth and development [56]. Its local accumulation in the pericycle of the root or hypocotyl triggers asymmetric cell divisions, as a first morphological hallmark of, respectively, lateral or adventitious root development [57,58]. In the root, the auxin homeostasis and signaling mechanisms that control lateral root spacing are well characterized [59,60], while this knowledge is lacking for AR induction in the hypocotyl [61]. Recently, the auxin co-receptors TIR1 and AFB2, their Aux/IAA counterparts IAA6, IAA9 and IAA17 [61], and downstream transcription factors ARF6, ARF8 [62] and the ARF7/ARF19 pair [63], have been implicated in AR formation. Light is an important regulator of AR formation. However, its effects on AR formation are somewhat ambiguous. It has been hypothesized that light stimulates AR formation via activating ARF6/8 [64], which was corroborated by the direct interaction between CRY1 and PHYB with ARF6 and ARF8 [65]. However, this was contradicted by Li et al. (2021) who showed that light, through PHYB, suppresses AR formation by stabilizing Aux/IAA proteins and by suppressing ARF7/19 activity [7]. These seemingly conflicting data illustrate the complexity by which light modulates auxin activities to control AR development.

A dark period prior to light treatment has been shown to enhance the rooting capacity of many plant species [66,67]. Insight into the genetic factors involved in light control of rooting is of primordial importance to crop propagation. Moreover, there are seemingly contradicting reports on how light and darkness affect adventitious root formation. The purpose of this study was to establish a mechanistic framework of light-regulated AR formation. Therefore, we used etiolated *Arabidopsis thaliana* seedlings as a well-established genetic model to explore how different light signaling and photomorphogenesis pathways control AR formation. In addition to the regulation of hypocotyl adventitious rooting capacity by the COP9 signalosome (CSN), SPA1, 2 and 3, and PIFs, we identified the transcription factor HY5 as a strong inhibitor of rooting competence. Together, these findings clearly position key regulators of light signaling and photomorphogenesis in the context of light-controlled AR induction. Based on our observations and known target processes of the respective components, we propose that the transition from dark to light triggers auxin redistribution and auxin sensitivity changes that result in AR inductive auxin activity maxima.

## 2. Results

### 2.1. Hypocotyl Adventitious Root Initiation Is Triggered by De-Etiolation

Light sensing triggers photomorphogenesis in dark-grown seedlings [68]. At the level of the hypocotyl, this includes the arrest of elongation [69], apical hook opening [70] and induction of adventitious roots (AR) [36]. The role of light and the process of photomorphogenesis on AR formation is, however, the least well understood. Therefore, we chose to study this in *Arabidopsis thaliana*, in which many mutants in light signaling and photomorphogenesis are readily available. The exposure of etiolated seedlings to light induced AR formation in *Arabidopsis thaliana* hypocotyls [71,72], while no AR appeared in the hypocotyls of seedlings that were germinated in the light (Figure 1a,b). Light-grown seedlings did not show indications of cell divisions or arrested primordia, which was visualized upon tissue clearing in the etiolated seedlings exposed to light (Figure 1c–e, Appendix A). An average of 1.487 AR primordia (ARP) were observed in hypocotyls of the etiolated seedlings transferred to light conditions (Figure 1f). Etiolation is therefore a critical precondition for light-dependent AR induction in *Arabidopsis thaliana* hypocotyls.

### 2.2. AR Formation in Etiolated Seedlings Is Stimulated by Blue, but Not by Red Light

Next, we aimed to characterize the effects of different light conditions on AR formation. We found that etiolated seedlings transferred into the light produced more AR than those kept in the dark (Figure 2). Continuous darkness, however, did not prevent AR induction, and AR primordia were observed in cleared hypocotyls,, indicating that light is not essential, yet enhances the capacity to form AR (Figure 2a,c). To determine the sensitivity to different light wavelengths, etiolated seedlings were exposed to white, blue, red and far-red LED light. The transfer of three days etiolated seedlings into the light strongly suppressed hypocotyl elongation with blue light showing the strongest inhibition (Figure 2a,b). Blue and white light stimulated AR formation, whereas red and far-red did not show a significant increase in the number of AR compared to seedlings kept in darkness (Figure 2c). The elongation and root induction response to blue light illumination was consistent with a previous report that identified blue light as a strong inducer of hypocotyl AR formation [9].

### 2.3. Blue Light Signaling Contributes to AR Formation

Since blue light is perceived by cryptochromes (*cry1* [73], *cry1cry2* [74]) and phototropins (*phot1phot2* [75]), we analyzed the contributions of these receptors to hypocotyl elongation and AR formation. Hypocotyls elongated in the dark (Figure 2b and Figure 3e), and this was suppressed by white (Figure 2b) and blue (Figure 2b and Figure 3e) light as expected. This suppression was significantly less in *cry1, cry1cry2* and *phot1phot2* mutants (Figure 3b). In the dark, AR formation in the photoreceptor mutants did not differ from the wild type controls (Figure 3d,f). In white light, ARP formation was strongly reduced in *cry1* and *cry1cry2* (Figure 3a,c), suggesting that blue light receptors are more important for AR formation than red and far-red signaling. Therefore, we explored in more detail the blue light signaling components (Figure 3). In white light, the cryptochrome mutants *cry1* and *cry1cry2* had longer hypocotyls than WT (Ler) and *phot1phot2* (Figure 3a,b), while AR formation in all tested blue light receptors was significantly reduced (Figure 3a,c). In blue light, a similar response was observed, except for the *phot1phot2* mutant that was responsive to blue light by inhibiting hypocotyl elongation (Figure 3d,e) and stimulating AR formation (Figure 3d,f). These data suggest a prominent role for CRY in hypocotyl elongation and AR formation, whereas PHOTs are not required for elongation and play only a minor role in AR formation.

### 2.4. CSN Subunits Play Differential Roles in AR Initiation

Since de-etiolation by white and blue light increased the ARP number, we asked whether activation of photomorphogenesis was sufficient for AR induction. To test this, constitutively photomorphogenic mutants of the COP9 signalosome were analyzed. Constitutive photomorphogenesis mutants produce short hypocotyls in the dark [76,77] and since hypomorphic alleles have been identified, namely, *csn2-5* [78], *csn3-3* [79], *csn5a-1* [80], *csn5a-2* [80] and *csn5b-1* [80], these were analyzed. These weak alleles did not show the short hypocotyl phenotype and instead, with the exception of *csn3-3*, hypocotyls were longer than the corresponding WTs (Figure 4a,b).

Previously, a weak allele of *CSN4* (*csn4-2035*) was identified as a suppressor for excessive AR production in the auxin-overproducing *superroot2-1* (*sur2-1*) mutant [36]. The *csn2-5* and *csn3-3* mutants formed fewer ARP than WT. The full and partial *CSN5A* knock-out alleles, *csn5a-1* and *csn5a-2*, respectively [80], hardly formed any ARPs. In contrast, the full knock-out in *CSN5B* (*csn5b-1*) [80] formed significantly more ARPs than WT (Figure 4a,c). This antagonistic function of CSN was also observed previously and reflects the differential contributions of the CSN5 subunits in AR initiation [36]. Taken together, these results suggest that activation of photomorphogenesis is not sufficient for AR stimulation.

### 2.5. COP1/SPA Complex Plays a Role in Dark-Light-Induced AR Initiation

The suppression of photomorphogenesis in dark-grown *Arabidopsis*
*thaliana* seedlings also requires, next to COP1, the activity of members of the SUPPRESSOR OF PHYA-105 (SPA) kinase family [81]. SPA kinases interact with and activate COP1 [82,83], forming a complex that promotes ubiquitination and degradation of ELONGATED HYPOCOTYL5 (HY5) to repress photomorphogenesis in the dark. Knocking out this COP1/SPA pathway is lethal or leads to developmental defects including dwarfism and early flowering [84], precluding the assessment of post-embryonic processes such as AR development. Therefore, we focused on mutants of the *SPA* family [22] that can complete the life cycle [82].

The hypocotyl of etiolated *spa1-100* [85], *spa2-2* [86] and *spa3-1* [87] was slightly longer than WT (Figure 5a,c), consistent with *SPA* involvement in shade-avoidance [88]. In contrast, the *spa1-3spa2-1spa3-1* (*spa1/2/3*) triple mutant [82] had a slightly shorter hypocotyl (Figure 5a,b). The moderate hypocotyl elongation defect in *spa1/2/3* is likely due to the presence of sucrose in the medium [82].

We then examined the dark-light transition-induced AR formation. While the *spa1-100*, *spa2-2* and *spa3-1* single mutants formed similar numbers of AR as Col-0, the *spa1/2/3* triple mutant did not form AR (Figure 5a,b). In the dark, COP1/SPA stimulates the proteolysis of bZIP transcription factors LONG HYPOCOTYL5 (HY5) and HY5 HOMOLOGUE (HYH), and in the light they are stabilized, activating light-dependent gene expression and photomorphogenesis [89]. Consistent with this dark-light regulatory system, *hy5* and *hyh* hypocotyls were longer than WT (Figure 5a,b) and formed more AR (Figure 5a,c). These results affirm that photomorphogenesis primarily suppresses AR formation.

### 2.6. Skotomorphogenesis PIF Factors Are Required for AR Formation

Phytochrome-Interacting Factors (PIFs) are basic helix-loop-helix (bHLH) transcription factors (TFs) that accumulate in the dark to promote skotomorphogenesis [90,91] and act antagonistically to HY5 in light responses [41,92,93]. To determine whether the etiolation factors PIF control AR formation, phenotypic responses in dark–to-light growth conditions were analyzed for single mutants *pif1-1*, *pif3-7*, *pif4-2* and *pif5-3*,and the multiplex mutant *pifQ* [42]. The single mutants developed slightly longer hypocotyls than the wild type (Figure 6a,b). In contrast, *pifQ* had shorter hypocotyls (Figure 6a,b), consistent with PIF functional redundancy in suppressing photomorphogenesis [94,95].

Similarly, the single knock-out mutants showed moderate to no changes in AR formation, while *pifQ* formed very few or no ARP four days after transfer into white light (Figure 6a,b). These results indicate that etiolation-induced AR competence depends on PIFs.

## 3. Discussion

### 3.1. Light Has Contrasting Effects on AR Formation

Light plays a pivotal role in plant growth and development [96], with other environmental factors [97]. Darkness and low light conditions induce etiolation and light triggers photomorphogenic growth. The transition from dark to light causes the arrest of hypocotyl elongation, opening of the apical hook, activation of the shoot meristem, greening and the induction of adventitious roots [8,98,99,100]. While many of these developmental processes have been studied intensively, AR induction is less well understood, and moreover, light has been reported to both stimulate and inhibit rooting [7].

The notion that photomorphogenesis and light signaling inhibit AR formation comes from the inhibition of AR formation in light-grown seedlings (Figure 1b). Mutants that express (partial) photomorphogenesis characteristics in the dark, e.g., *cns*, *pifQ* and *spa1/2/3* triple (Figure 5), and mutants expressing a constitutive active PHYB [7], show a severely reduced capacity of AR formation. Consistently, the incapacity to execute photomorphogenesis in the *hy5* mutant [101] resulted in increased AR formation.

This complementarity leads to the interpretation that skotomorphogenesis is associated with increased AR competence. In *Arabidopsis*
*thaliana* hypocotyls, there seems to be an optimal etiolation time for AR induction, after which the light-induced AR production becomes less efficient [7]. This is reflected in the lack of AR formation in *pifQ* mutants that are defective in the major skotomorphogenesis transcription factors (Figure 6). Consistent with enhanced AR competence in etiolated seedlings, a dark treatment is often used to improve AR formation of cuttings as shown for *Petunia hybrida* and *Prunus avium* [66,67], and in micropropagation of various horticulture and tree crops as in, e.g., *Acacia mangium* and *Malus domestica* (Borkh.) Likhonos [102,103]. Therefore, this improved AR formation might be explained by a partial reversal of the photomorphogenetic state during the dark treatment, which alleviates the suppression of AR formation.

The rooting responses of skotomorphogenesis and photomorphogenesis *Arabidopsis*
*thaliana* mutants illustrate how darkness installs a state of AR competence and light reduces AR formation. Light, however, stimulates AR formation when applied to etiolated seedlings within a limited time period [7]. Blue light and red light, but not far-red light, are AR inductive signals in etiolated hypocotyls [9]. In our experiments, stronger stimulation of AR formation by blue light than with red light was observed, presumably because we used higher intensities than in previous studies. Light intensity may therefore also be a factor determining the capacity to induce AR formation in etiolated seedlings.

The AR inducing effect by light has been proposed to be related to the activation of photosynthesis [104]. However, in our experiments, the blue-light effect was entirely dependent on the blue light receptors CRY1 and CRY2, and partially on PHOT1/PHOT2, indicating that photosynthesis played only a minor role. Photosynthesis may however contribute to AR formation during further development of the AR primordia by activating the small GTPase ROP2 and TOR kinase [105], leading to AR formation in potato [106].

In conclusion, our data show that light acts as an inhibitor of AR competence when applied to light-grown seedlings, while it functions as an AR stimulus in dark-grown seedlings.

### 3.2. Auxin Plays a Central Role in the Dual Effect of Light on AR Formation

Auxin is the central signaling hormone for AR induction. Its accumulation in the pericycle activates asymmetric cell division and subsequent AR organogenesis [61]. Light is therefore expected to control AR formation via auxin. One of the major molecular differences between light and darkness lies in the antagonism between PIF and HY5 transcription factors that are stable in darkness and light, respectively. A direct target of PIFs are auxin biosynthesis genes of the *YUCCA* family [107,108]. In the dark, PIFs stimulate auxin biosynthesis and act cooperatively with the auxin ARF6 and brassinosteroid BZR1 transcription factors [109]. ARF6 mediates auxin regulation of AR induction [34,62] and brassinosteroids stimulate AR formation in the auxin-overproducing *sur2-7* mutant [110]. A similar cooperation between PIFs, ARF6 and BZR1 may control AR competence in the dark.

In light, PIFs are degraded and HY5 is stabilized, activating photomorphogenesis-related genes [111]. We found that dark-grown *pifQ* and *hy5* mutants form, respectively, less and more ARs upon transfer to light. One possible explanation for this phenomenon can be found in their respective known targets. While PIFs enhance auxin levels in the dark [112], HY5 suppresses auxin signaling in the light by inducing Aux/IAA signaling repressors *SLR/IAA14* and *AXR2/IAA7* [113] and suppressing *YUCCA9* [114]. Consistently, it was recently found that extended darkness leads to the formation of ARs independent of a light stimulus [7]. This underlines the central role of auxin and darkness in determining the competence to form AR.

Next to auxin biosynthesis, auxin receptivity also determines AR formation. This follows from the observed reduction in AR in *csn* mutants (Figure 4) [36], which display auxin resistance [31] due to a direct impact on auxin co-receptor SCF^TIR1/AFB^ activity [115] via its role in deconjugation of NEDD8/RUB1 of the CUL1 subunit in SCF complexes. The nuclear localization and HY5-degrading activity of COP1 in the dark depends on CSN [25], causing the constitutive photomorphogenic phenotype of *csn* mutants, revealing that CSN contributes to AR competence in etiolated seedlings possibly via two separate signaling routes.

In blue light, auxin signaling is mediated via cryptochromes and phototropins. CRY1 interacts with the transcription factors PIF4 and PIF5 [43,116,117,118] and with SPA1 to suppress COP1-dependent degradation of the transcription factor HY5 [119,120]. In addition, CRY1 counteracts the association of TIR1 and AUX/IAAs [121] and represses DNA binding of ARF6 and ARF8 [65]. Cryptochrome-mediated blue light signaling thereby suppresses auxin signaling, and thus AR formation, in the hypocotyl. However, we found that CRY1 is required for blue light-induced AR formation (Figure 3). CRYs, therefore, also positively influence auxin signaling in the context of AR formation, a process that awaits elucidation.

A hint for the mechanism by which blue light might regulate AR induction comes from *phot* mutant analysis (Figure 3). PHOT1-mediated blue light perception in AR formation was proposed to result in the modulation of auxin transport via PIN3 activity [9]. PHOT1 also inhibits the auxin transporter ABCB19, changing auxin flux from the shoot to the hypocotyl [122]. This is consistent with a model in which blue light affects auxin flux from the shoot to the hypocotyl and modulates PIN3-mediated auxin transport to the pericycle to induce AR formation.

### 3.3. Model for Dual Role of Light in AR Formation

Light and darkness have contrasting effects on auxin homeostasis and signaling. Here, we propose that the switch from darkness to light results in a transient and local accumulation of auxin in the pericycle that reconciles both effects on the induction of AR.

During skotomorphogenesis, auxin biosynthesis rates are high to stimulate rapid growth and to maintain the apical hook [123,124] (Figure 7a). The transfer to light and activation of photomorphogenesis causes an important relocalization of auxin and suppression of auxin signaling in the entire hypocotyl, resulting in an arrest of hypocotyl growth and the induction of apical hook opening (Figure 7b). We anticipate that the redistribution of auxin, further facilitated via blue light effects on auxin transport, results in local auxin maxima in the pericycle that are sufficiently strong to trigger asymmetric division and AR formation. Over time, light signaling and photomorphogenesis install auxin resistance at the level of the signaling machinery, precluding further AR induction (Figure 7c). Future studies to test this hypothesis require the analysis of temporal changes of auxin levels and transport in the *Arabidopsis*
*thaliana* hypocotyl.

## 4. Materials and Methods

### 4.1. Plant Materials and Growth Conditions

The *phot1phot2* [75], *csn3-3* [79], *csn5a-1* [80], *csn5a-2* [80], *csn 5b-1* [80], *spa1-100* [85], *spa2-2* [86], *spa3-1* [87], *spa1-3spa2-1spa3-1* (*spa1/2/3*) triple mutant [82], *pif1-1*(SAIL_256G07), *pif3-7* (CS66042), *pif4-2* (SALK_140393), *pif5-3* (CS66044) and *pifQ* (*pif1-1pif3-3pif4-2pif5-3*) [42] used in this study are in the Columbia-0 (Col-0) background. The *cry1* [73], *cry1cry2* [74] and *csn2-5* [78] are in the Landsberg *erecta* (Ler) background. The *hy5* [125] and *hyh* [89] are in the Wassilewskija (Ws) background. The single pif mutants were obtained from NASC (The Nottingham Arabidopsis Stock Centre). Homozygous, genotyped lines were selected after two complete growth cycles (primers in Appendix A). Wild type Arabidopsis thaliana seeds were collected at the same time for assays. Seeds were surface sterilized and sown on 1/2 MS agar vertical plates (1.5 g/L MS, 0.5% (*w*/*v*) sucrose, 0.05% (*w*/*v*) MES and 0.8% (*w*/*v*) agar, pH 5.7). Plates were incubated for 4 d at 4 °C in dark for stratification. Plants were germinated by 8 h incubation in the light (22 °C, 70 µmol/m^2^s) before incubated for 3 d in the dark to induce hypocotyl elongation as described [71]. Well-elongated seedings were transferred to 1/2 MS agar vertical plates and further grown on these plates for 4 d in a growth chamber at 70% relative humidity and 22 °C, with 16 h/8 h light/dark cycles (150 µmol/m^2^s).

### 4.2. Light Sources

For light-response experiments, seedlings were grown in darkness for 3 days at 22 °C with 16 h/8 h light/dark cycles after stratified 4 days at 4 °C in dark and followed by stimulation under blue light (peak: 470 nm, half band width: 30 nm, 59.75 µmol/m^2^s), red light (peak: 660 nm, half band width: 20 nm, 41.62 µmol/m^2^s), far-red light (peak: 740 nm, half band width: 25 nm, 1.32 µmol/m^2^s) or darkness for 4 days for further analysis. The light condition was applied by Philips Greenpower light-emitting diode (LED) chambers, which are spectrally controllable by Philips GrowWise control system. Total incident light intensity and the spectral distributions of the different light sources were measured using a spectroradiometer (SS-110, Apogee Instruments, Logan, UT, USA).

### 4.3. Hypocotyl Phenotypic Analysis

Images of seedlings on vertical plates were taken with D7000 Nikon camera, AF-S VR Micro-Nikkor 105 mm f/2.8 G IF-ED lens, followed by manual analysis using the ImageJ software plugin (http://www.imagescience.org/meijering/software/neuronj/, accessed on 26 March 2021) [126] for hypocotyl length measurement.

### 4.4. Preparation and Observation of Cleared Seedings

For light microscopy, plant seedlings were cleared with methanol and NaOH and mounted as described in [127]. Seedlings were harvested and fixed in acetone (90%) overnight at 4 °C. After fixation, seedlings were transferred to 0.5 M phosphate buffer for 30 min at 37 °C, followed by 45 min in clearing solution I (0.24 N HCl in 20% methanol) at 60 °C and 15 min in Clearing Sol II (7% NaOH in 60% EtOH) at room temperature. Seedings were then rehydrated sequentially with ethanol series (40, 20 and 10%) at room temperature for 5 min each step and infiltrated for at least 1 h with 5% ethanol in 25% glycerol. Subsequently, the entire seedlings were mounted in 50% glycerol (our trick to keep the seedlings straight: putting the cleared seedlings on the coverslip rather than on the microscopy slide) and adventitious root primordia were inspected by BX51 microscope (Olympus, Tokyo, Japan) using differential interference contrast (DIC) optics.

### 4.5. Statistical Analyses

Statistical analysis was performed using GraphPad Prism 8. One way ANOVA and post-hoc tests were used to assess differences between the mutant lines (*p* < 0.05). Nonpaired Student *t*-test was used for two group comparison (Figure 1f). All mutants in Figure 5, Figure 6, Figure 7 were analyzed in the same runs, resulting in duplication of the WT data, upon splitting the data up over different figures. For all comparisons, at least three independent experiments were performed, each with more than 20 seedlings for mutant lines.

## 5. Conclusions

Despite AR formation being intimately connected to light signaling and photomorphogenesis, two heavily studied physiological processes, the mechanism by which these processes converge on AR formation has remained elusive. Here, we analyzed the AR competence of different mutants in key positions in these pathways, providing a first outline of the genetic framework that controls light-controlled AR formation. Integrating these findings with known targets of these components led us to propose the above model based on the contrasting effects of light and darkness on auxin signaling and homeostasis. Validation of this model will require mapping and modeling of the spatio-temporal characteristics of auxin signaling and homeostasis in tissues relevant for dark to light-induced AR formation, and how this is affected in the used light signaling and photomorphogenesis mutants. A deeper understanding of the molecular mechanism of light-regulated AR formation may inspire novel strategies for improving AR formation during clonal propagation of crop and ornamental species.

## Figures and Tables

**Figure 1 ijms-23-05301-f001:**
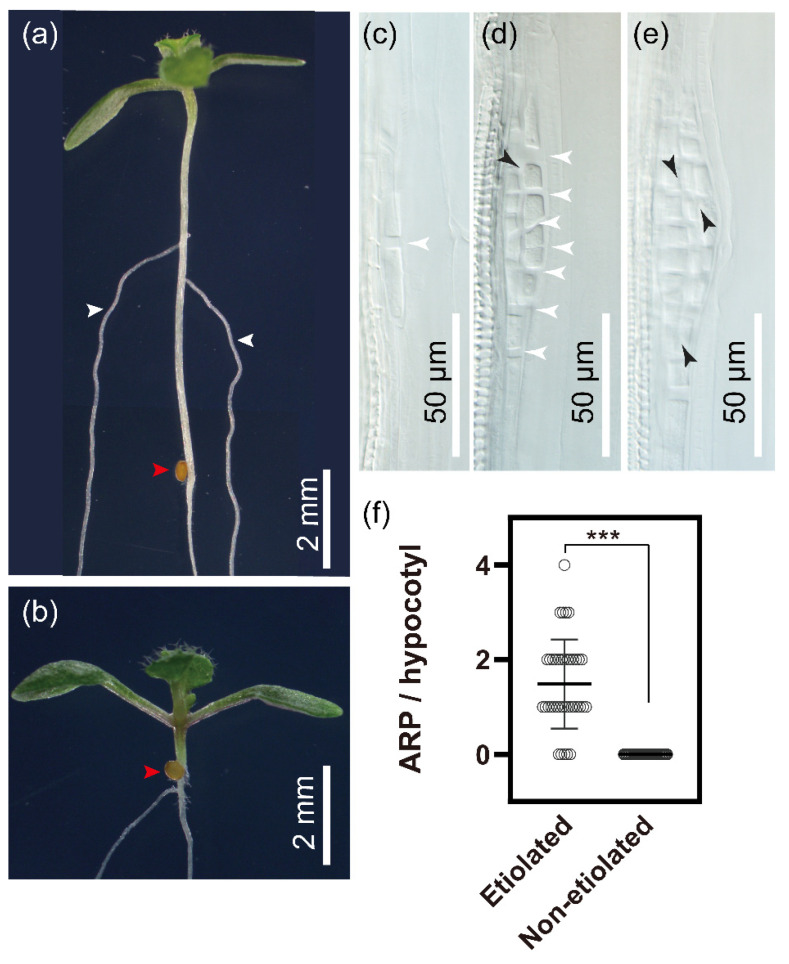
Adventitious root formation in etiolated and non-etiolated hypocotyls of *Arabidopsis*
*thaliana*. (**a**) AR formation upon exposure to light of etiolated (7 days) Col-0, (**b**) no hypocotyl AR are formed in light grown (7 days) Col-0, (red arrowhead points to the hypocotyl root junction, scale bar: 2 mm). (**c**) Stage I ARP, (**d**) stage II ARP and (**e**) stage III ARP in cleared hypocotyls from 2 days de-etiolated seedlings. White arrowheads point to cell walls from anticlinal or oblique cell divisions; black arrowheads point to cell walls from periclinal divisions; scale bar: 50 μm. (**f**) Adventitious root number per hypocotyl in de-etiolated and non-etiolated Col-0 seedlings. AR were quantified after 7 days in the light. The data represented as means ± SD (etiolated, *n* = 39; non-etiolated, *n* = 37). (***: *p* ≤ 0.001, unpaired Student’s *t*-test).

**Figure 2 ijms-23-05301-f002:**
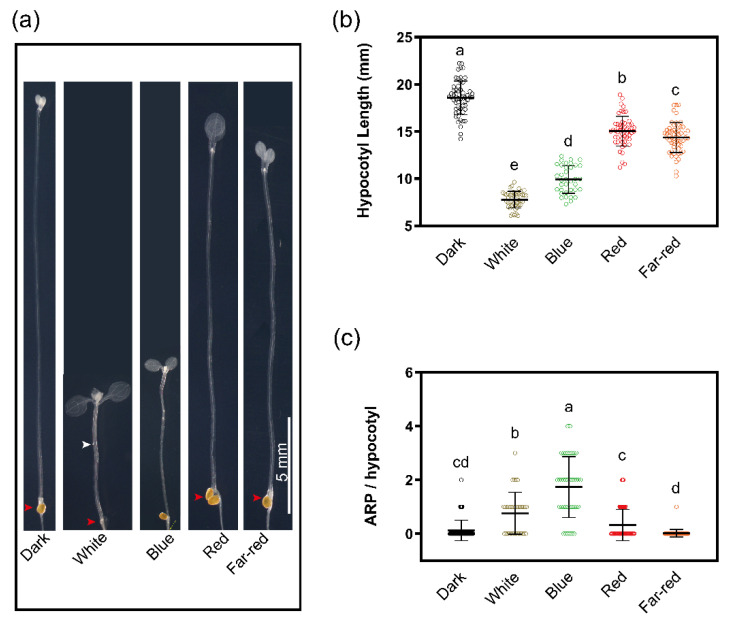
Hypocotyl elongation and adventitious root formation under different light sources. (**a**) Representative hypocotyl images of three-day-old etiolated WT (Col-0) seedlings that were illuminated with different LED light for 4 days. White arrowheads indicate ARPs, the red arrowhead points to the hypocotyl root junction. (**b**) Quantification of the hypocotyl length of the seedlings as in (**a**). (**c**) Quantification of the number of ARP in the hypocotyl of the seedlings as in (**a**). The data in (**b**,**c**) are represented as mean values ± SD, *n*-dark = 56; *n*-white = 34; *n*-blue = 38; *n*-red = 53; *n*-far-red = 51). Different letters indicate a significant difference at *p* ≤ 0.05 (ANOVA and LSD post hoc analysis). Scale bar: 5 mm. All images were taken at the same magnification.

**Figure 3 ijms-23-05301-f003:**
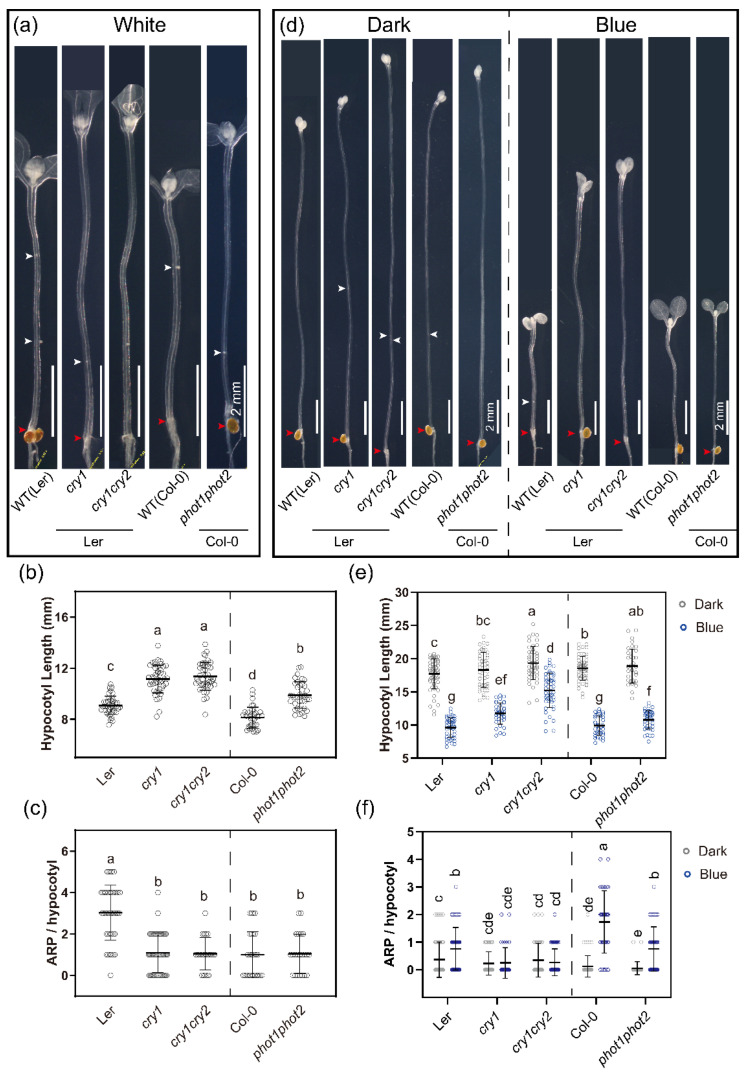
Blue light photoreceptors promote adventitious root induction. (**a**) Representative images of WT (Ler), *cry1* (Ler), *cry1cry2* (Ler), WT (Col-0) and *phot1phot2* (Col-0) cleared hypocotyls grown in white light for 4 days after 3 days etiolation. Scale bar: 2 mm. (**b**,**c**) Quantification of the (**b**) hypocotyl length and (**c**) ARP of seedlings as in (**a**). (**d**) Representative hypocotyl images of three-day-old etiolated seedlings illuminated with blue light for 4 days. White arrowheads indicate ARPs, the red arrowhead points to the hypocotyl root junction. (**e**,**f**) Quantification of the (**e**) hypocotyl length and (**f**) number of ARP. The data in b and c (Ler, *n* = 40; *cry1*, *n* = 45; *cry1cry2*, *n* = 42; Col-0, *n* = 37; *phot1phot2, n = 37*), and in e and f (Ler, *n* = 60, 50; *cry1*, *n* = 44, 36; *cry1cry2*, *n* = 46, 52; Col-0, *n* = 56, 38; *phot1phot2, n = 35, 45* for dark and blue light, respectively) represent mean values ± SD. Different letters indicate a significant difference at *p* ≤ 0.05 (ANOVA and LSD post hoc analysis). Scale bars in all pictures = 2 mm.

**Figure 4 ijms-23-05301-f004:**
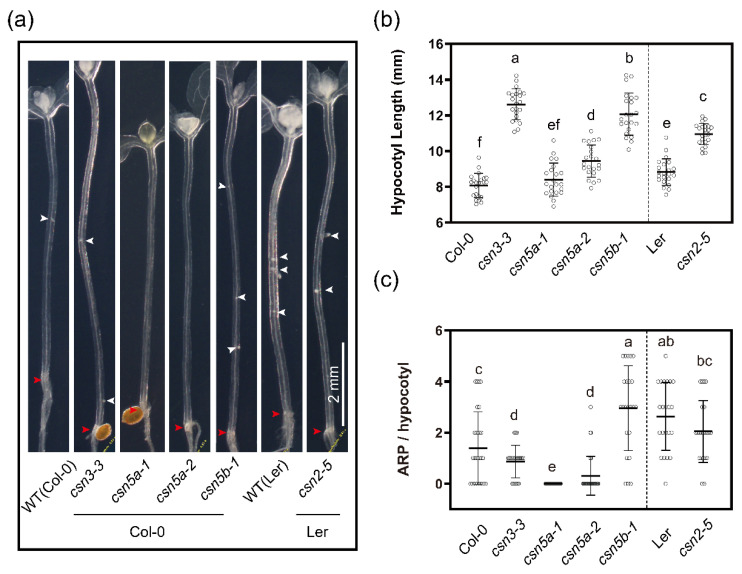
Light-induced ARP formation is impaired in the CSN subunits mutants. (**a**) Representative images of cleared hypocotyls of WT (Col-0), *csn3-3* (Col-0), *csn5a-1* (Col-0), *csn5a-2* (Col-0), *csn5b-1* (Col-0), WT (Ler) and *csn2-5* (Ler) grown in the light for 4 days after 3 days etiolation in the dark. (**b**,**c**) Quantification of the (**b**) hypocotyl length and (**c**) number of ARP of the seedlings as shown in (**a**). The data are presented as mean values ± SD, Col-0, *n* = 27; *csn3-3*, *n* = 22; *csn5a-1*, *n* = 26; *csn5a-2*, *n* = 29; *csn5b-1*, *n* = 23; Ler *n* = 22 and *csn2-5, n* = 22. White arrowheads indicate ARPs, the red arrowhead points to the hypocotyl root junction. Different letters indicate a significant difference at *p* ≤ 0.05 (ANOVA and LSD post hoc analysis). Scale bar: 2 mm. All images are taken at the same magnification.

**Figure 5 ijms-23-05301-f005:**
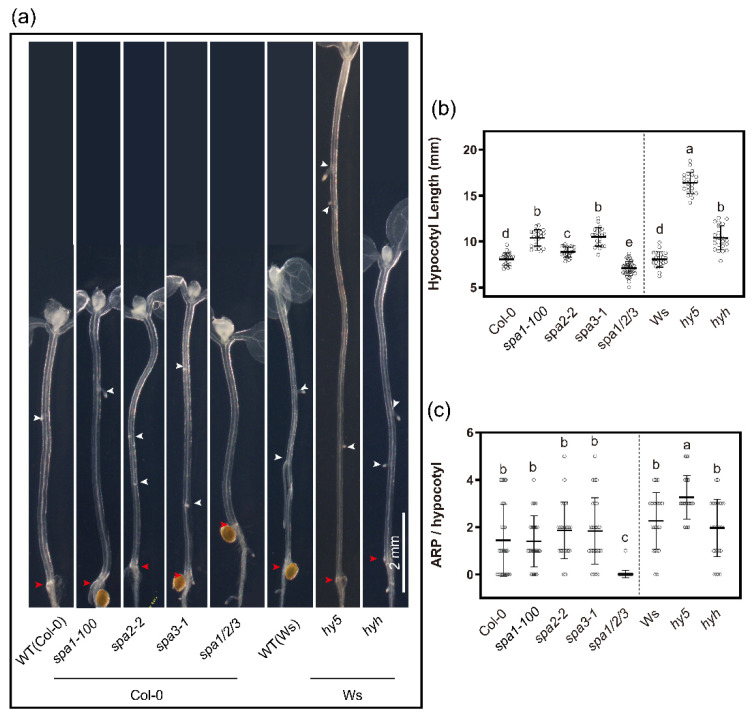
Hypocotyl elongation and adventitious root formation phenotypes of *spa1,2,3* single and triple mutants, *hy5* and *hyh*. (**a**) Representative images of WT (Col-0), *spa1-100* (Col-0), *spa2-2* (Col-0), *spa3-1* (Col-0), *spa1/2/3* triple (Col-0), WT (Ws), *hy5* (Ws) and *hyh* (Ws) mutant cleared hypocotyl grown in the light for 4 days after 3 days etiolation. White arrowheads indicate ARPs, the red arrowhead points to the hypocotyl root junction. (**b**,**c**) Quantification of the (**b**) hypocotyl length and (**c**) ARP of the seedlings as in (**a**). The data in (**b**,**c**) are represented as mean values ± SD (Col-0, *n* = 27; *spa1-100*, *n* = 25; *spa2-2*, *n* = 22; *spa3-1*, *n* = 25; *spa1/2/3* triple *n* = 40; Ws, *n* = 22; *hy5 n* = 23 and *hyh, n* = 26). Different letters indicate a significant difference at *p* ≤ 0.05 (ANOVA and LSD post hoc analysis). Scale bar: 2 mm. All images were taken with the same magnification.

**Figure 6 ijms-23-05301-f006:**
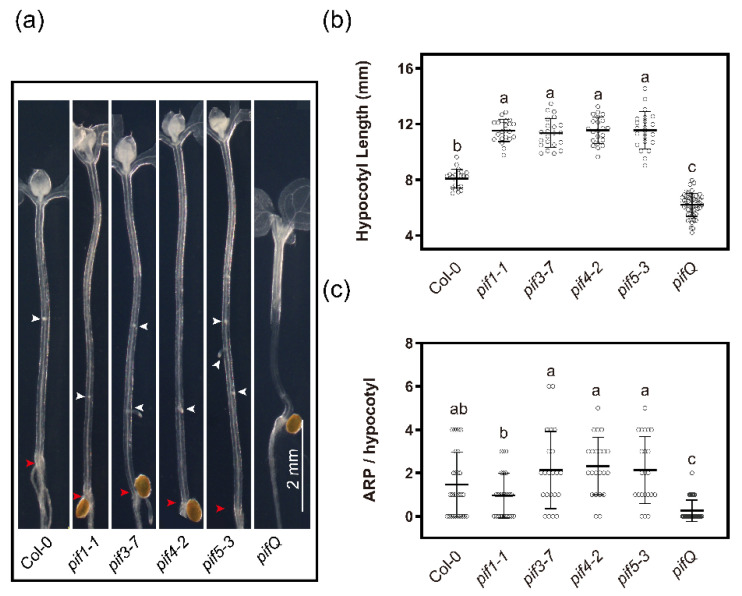
Hypocotyl elongation and rooting response of Phytochrome-Interacting Factor (PIF) mutants. (**a**) Representative images of WT (Col-0), *pif1-1* (Col-0), *pif3-7* (Col-0), *pif4-2* (Col-0), *pif 5-3* (Col-0) and *pifQ* (Col-0) cleared hypocotyl grown in the light for 4 days after 3 days etiolation. White arrowheads indicate ARP, the red arrowhead points to the hypocotyl root junction. (**b**,**c**) Quantification of the (**b**) hypocotyl length and (**c**) ARP of the seedlings as in (**a**). The data in (**b**,**c**) are means ± SD obtained from mutant lines (Col-0, *n* = 27; *pif1-1*, *n* = 24; *pif3-7*, *n* = 22; *pif4-2*, *n* = 22; *pif 5-3*, *n* = 22 and *pifQ*, *n* = 46). Different letters indicate a significant difference at *p* ≤ 0.05 (ANOVA and LSD post hoc analysis). Scale bar: 2 mm. All images were taken at the same magnification.

**Figure 7 ijms-23-05301-f007:**
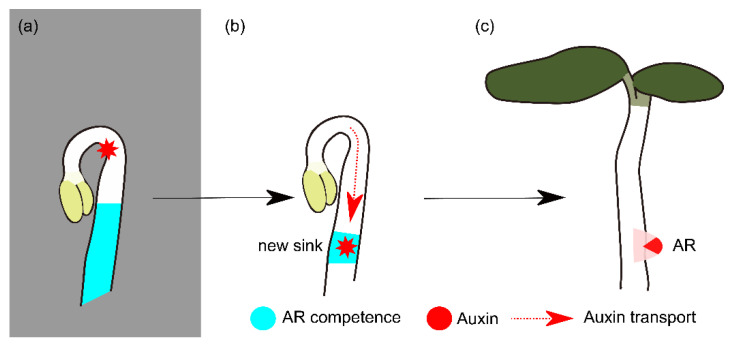
Model for light-induced adventitious root formation. (**a**) In darkness, skotomorphogenetic growth is governed by an auxin maximum that maintains the apical hook, rapid elongation growth and high adventitious rooting competence. (**b**) Light signaling antagonizes auxin activities that control apical hook maintenance and elongation growth by activating auxin transport and dampening auxin signaling, resulting in the formation of a new auxin sink at the site of AR initiation. (**c**) Photomorphogenesis is complete. The apical hook has opened, cotyledons are expanded and green, and the adventitious roots develop further. At this point the competence for AR formation is strongly reduced, and no new ARs are initiated.

## Data Availability

Raw data supporting reported results are stored on a server in accordance to rules outlined by the data management plan of Ghent University (www.ugent.be/en/research/datamanagement/before-research/datamanagementplan.htm, accessed on 6 April 2022).

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
