# Peer review of "Genetic Dissection of Light-Regulated Adventitious Root Induction in Arabidopsis thaliana Hypocotyls"

_ijms, 2022, doi:10.3390/ijms23105301_

Round 1
Reviewer 1 Report
The authors propose a manuscript titled “Genetic dissection of light-regulated adventitious root induction in Arabidopsis hypocotyls”. The article is original, well structured and written. The study takes into consideration and highlights an interisting topic on on photomorphogenic responses of etiolated seedlings that include the inhibition of hypocotyl elongation and opening of the apical hook andthe fact that How light signaling controls adventitious rooting is less knowed. In particular the blue light receptors CRY1 and PHOT1/PHOT2 are key elements contributing to the induction of AR formation in response to light. The results provide a genetic framework that explains high and low AR competence in the dark, and light respectively. The authors propose that light-induced auxin signal dissipation generates a transient auxin maximum that explains AR induction by a dark to light switch. I think that after few suggestions, the work may be published.
Title please with scientific name consider also the author that describe the genus, in this way: Arabidopsis Heynh.
- Introduction. few observations. Please complete or add reference in the suggested way.
- Lines 37-43. Light, as many other environmental factors, is of fundamental importance for the autotrophic growth of plants (choose reference). Therefore, it is not surprising that plants tune their body plan through the correct positioning of aerial organs or to accelerate elongation to outgrow competing plants to optimally capture and use the available light [1,2]. In fact, light triggers a major developmental transition just after germination (choose reference). Seedlings that germinated in the dark, grow almost exclusively by elongation in their search for light, which in turn signals they have reached the soil surface (choose reference). This is referred to as skotomorphogenesis (choose reference).”;
- Lines 82-83. The most prominent phenotype of skotomorphogenic development is hypocotyl elongation and apical hook formation (choose reference);
- Lines 90-92. Its local accumulation in the pericycle of the root or hypocotyl triggers asymmetric cell divisions, as a first morphological hallmark of respectively lateral or adventitious root development (choose reference);
- Line 104. Arabidopsis in italic. Remember that the scientific name must be reported in italic. Please check whole document!
- Results and 3. Discussion
Well done, the figures are clear. Few suggestions
- Line 211. Arabidopsis in italic;
- Line 269. Light plays a pivotal role in plant growth and development (choose reference), with other environmental factors (Perrino et al. 2021);
- Lines 288-290. See my previous comment and correct accordingly:
- Prunus avium
- Acacia mangium
- Malus domestica (Borkh.) Likhonos
References to be added:
- Perrino, E.V.; Valerio, F.; Jallali, S.; Trani, A.; Mezzapesa, G.N. Ecological and Biological Properties of Satureja cuneifolia and Thymus spinulosus Ten.: Two Wild Officinal Species of Conservation Concern in Apulia (Italy). A Preliminary Survey. Plants 2021, 10, 1952. doi: 10.3390/plants10091952.
- Materials and Methods. Well done. few corrections
- Line 388. Arabidopsis thaliana (L.) Heynh.
- Line 425. Please version of ANOVA software used…
Conclusion
No conclusions? Why?
References
Please consider doi when is available and follow the guidelines of the journal
Author Response
ijms-1688256_Reply to reviewers:
Comments from the Reviewer #1 (R1):
The authors propose a manuscript titled “Genetic dissection of light-regulated adventitious root induction in Arabidopsis hypocotyls”. The article is original, well structured and written. The study takes into consideration and highlights an interisting topic on on photomorphogenic responses of etiolated seedlings that include the inhibition of hypocotyl elongation and opening of the apical hook andthe fact that How light signaling controls adventitious rooting is less knowed. In particular the blue light receptors CRY1 and PHOT1/PHOT2 are key elements contributing to the induction of AR formation in response to light. The results provide a genetic framework that explains high and low AR competence in the dark, and light respectively. The authors propose that light-induced auxin signal dissipation generates a transient auxin maximum that explains AR induction by a dark to light switch. I think that after few suggestions, the work may be published.
R1C1: Title please with scientific name consider also the author that describe the genus, in this way: Arabidopsis Heynh.
R1: We modified the title as: Genetic dissection of light-regulated adventitious root induc-tion in Arabidopsis thaliana hypocotyls
Introduction. few observations. Please complete or add reference in the suggested way.
R1C2: Lines 37-43. Light, as many other environmental factors, is of fundamental importance for the autotrophic growth of plants (choose reference). Therefore, it is not surprising that plants tune their body plan through the correct positioning of aerial organs or to accelerate elongation to outgrow competing plants to optimally capture and use the available light [1,2]. In fact, light triggers a major developmental transition just after germination (choose reference). Seedlings that germinated in the dark, grow almost exclusively by elongation in their search for light, which in turn signals they have reached the soil surface (choose reference). This is referred to as skotomorphogenesis (choose reference).”;
R2: we modified the text accordingly.
R1C3: Lines 82-83. The most prominent phenotype of skotomorphogenic development is hypocotyl elongation and apical hook formation (choose reference);
R3: we modified the text accordingly.
R1C4: Lines 90-92. Its local accumulation in the pericycle of the root or hypocotyl triggers asymmetric cell divisions, as a first morphological hallmark of respectively lateral or adventitious root development (choose reference);
R4: we modified the text accordingly.
R1C5: Line 104. Arabidopsis in italic. Remember that the scientific name must be reported in italic. Please check whole document!
R5: we modified the text accordingly.
- Results and 3. Discussion
Well done, the figures are clear. Few suggestions
R1C6: Line 211. Arabidopsis in italic;
R6: we modified the text accordingly.
R1C7: Line 269. Light plays a pivotal role in plant growth and development (choose reference), with other environmental factors (Perrino et al. 2021);
R7: we modified the text accordingly.
R1C8: Lines 288-290. See my previous comment and correct accordingly:
- Prunus avium
- Acacia mangium
- Malus domestica (Borkh.) Likhonos
R8: we modified the text accordingly.
References to be added:
- Perrino, E.V.; Valerio, F.; Jallali, S.; Trani, A.; Mezzapesa, G.N. Ecological and Biological Properties of Satureja cuneifolia and Thymus spinulosus Ten.: Two Wild Officinal Species of Conservation Concern in Apulia (Italy). A Preliminary Survey. Plants 2021, 10, 1952. doi: 10.3390/plants10091952.
- Materials and Methods. Well done. few corrections
R1C9: Line 388. Arabidopsis thaliana (L.) Heynh.
R9: we modified the text accordingly.
R1C10: Line 425. Please version of ANOVA software used…
R10: We have added additional information for the software in the text.
Conclusion
R1C11: No conclusions? Why?
R11: We have included a model in figure 7 that illustrates the hypothesis of new sink formation after a dark light switch. A conclusion is added.
References
Please consider doi when is available and follow the guidelines of the journal
Yes, we considered this.
Reviewer 2 Report
Manuscript "Genetic dissection of light-regulated adventitious root induction in Arabidopsis hypocotyls" is very interesting.
Authors used etiolated Arabidopsis seedlings to explore the role of light in the control of AR formation in the hypocotyl in greater detail. By analyzing mutants defective in light signaling and photomorphogenesis responses, they provide evidence for a profound negative effect of light signaling and photomorphogenesis on AR competence, while the dark and skotomorphogenesis define a state of heightened AR competence. Authors proposed that the transition from dark to light triggers auxin redistributions and auxin sensitivity changes that result in AR-inductive auxin activity maxima.
Quality of Figure 1f is very poor. Lack information about distribution of observed traits before statistical analysis.
Lack of analysis of relationships between observed traits.
4.5. Statistical Analyses: "Nonpaired Student t test was used for two group comparison." but in Lines 152, 181, 208, 232 and 263 is information about LSD Post-Hoc test. This is not the same. It requires improvement.
Paper needs minor revision.
Author Response
ijms-1688256_Reply to reviewers:
Comments from the Reviewer #2:
Manuscript "Genetic dissection of light-regulated adventitious root induction in Arabidopsis hypocotyls" is very interesting.
Authors used etiolated Arabidopsis seedlings to explore the role of light in the control of AR formation in the hypocotyl in greater detail. By analyzing mutants defective in light signaling and photomorphogenesis responses, they provide evidence for a profound negative effect of light signaling and photomorphogenesis on AR competence, while the dark and skotomorphogenesis define a state of heightened AR competence. Authors proposed that the transition from dark to light triggers auxin redistributions and auxin sensitivity changes that result in AR-inductive auxin activity maxima.
R2C1: Quality of Figure 1f is very poor. Lack information about distribution of observed traits before statistical analysis.
Lack of analysis of relationships between observed traits.
R1: We changed the figure with a higher quality image. Now the resolution has increased. About the statistics and the interactions between hypocotyl length and the number of AR. In the light (non-etiolated) seedlings there are no roots formed. Therefore, an analysis to correlate AR number and hypocotyl length is not going to reveal new insights into the relation between these two traits. We have investigated this question and correlated hypocotyl length with AR number in several mutants and found that there is not strict positive or negative relationship. Here, in this figure, the correlation is not the point we present, it merely illustrates that etiolation is suppression AR formation.
Plot: Linear regression models displaying no correlations between ARP number and hypocotyl length of de-etiolated Arabidopsis thaliana. The solid line is a linear regression model, dashed lines with green area are 95% confidence intervals. Regression equations, multiple R2, p-values and sample size are given. Individual symbols are individual seedlings, data are collected after the plants grown in the light for 4 days after 3 days etiolation.
R2C2:4.5. Statistical Analyses: "Nonpaired Student t test was used for two group comparison." but in Lines 152, 181, 208, 232 and 263 is information about LSD Post-Hoc test. This is not the same. It requires improvement.
Paper needs minor revision.
Reply: We have added additional information in the text for the Student t test comparison.
Reviewer 3 Report
Dear Authors,
The present study analyzes adventitious rooting under different light conditions in wild type and photomorphogenesis mutants. The research subject is interesting and brings scientific important data in the field, as it deals with a subject that is currently of great interest. Some changes of the manuscript should nevertheless be performed in order to improve its quality. Following specific changes should thus be performed:
Minor changes
I do not understand the meaning of highlighting some parts in yellow.
All scientific names of species or genera should be italic throughout the whole manuscript.
Major changes
Introduction: This section should contain informations regarding similar existing studies in literature and, in comparison, authors should emphasize the novelty and originality of their study. If these studies do not exist, please state and highlight your novelty and originality. In the part you describe the purpose of the study, you should state why the results obtained in this study may prove important. The purpose of the study is not clear, please add more informations and details. Another big problem in related to the species that are chosen for study. Arabidopsis is a genera of species. You need to mention on which exact species of the genus you have performed your studies on, as the genera contains several. Please offer a rationale for choosing these species and offer more details about them in the context of your study.
Discussions: Here you should emphasize novelty and originality of the present study once again. You compare your results with the ones obtained by other authors, but you need to highlight what you bring in novelty compared to these. Another observation is related to the fact that you cannot include conclusions in this section. Please revise accordingly.
Materials and methods: Not all methods have references. Are they completely novel? If not, please offer references.
Conclusions: You need to add them because your Discussions and Results sections are quite long and it is necessary to draw some important aspects regarding your findings in the study. Please offer interpretations of the findings in your study, alongside with perspectives.
All these suggested changes should be performed in order to bring further improvements to the manuscript.
Author Response
ijms-1688256_Reply to reviewers:
Comments from the Reviewer #3 (R1):
Dear Authors,
The present study analyzes adventitious rooting under different light conditions in wild type and photomorphogenesis mutants. The research subject is interesting and brings scientific important data in the field, as it deals with a subject that is currently of great interest. Some changes of the manuscript should nevertheless be performed in order to improve its quality. Following specific changes should thus be performed:
R3C1: Minor changes
I do not understand the meaning of highlighting some parts in yellow.
All scientific names of species or genera should be italic throughout the whole manuscript.
R1: We have added additional information to explain the highlighted text. Organism names are now in italic.
Major changes
R3C2: Introduction: This section should contain informations regarding similar existing studies in literature and, in comparison, authors should emphasize the novelty and originality of their study. If these studies do not exist, please state and highlight your novelty and originality. In the part you describe the purpose of the study, you should state why the results obtained in this study may prove important. The purpose of the study is not clear, please add more informations and details. Another big problem in related to the species that are chosen for study. Arabidopsis is a genera of species. You need to mention on which exact species of the genus you have performed your studies on, as the genera contains several. Please offer a rationale for choosing these species and offer more details about them in the context of your study.
R2: The abstract points out the novelty of our work. The capacity of a plant or plant tissue to form adventitious rooting is much higher after it has been exposed to darkness and then moved to the light. We identify the light signaling factors COP1, SPA kinases and PIFs as promoters and COP9 and HY5 as inhibitors of the rooting competence. Furthermore, we show that the increased competence is likely associated with a light induced auxin signaling creating a transient auxin maximum that explains the AR induction. These are the novel findings. In the introduction we provide background information on light signaling and auxin signaling so the reader gets a brief overview of the state or art on these two topics.
R3C3: Discussions: Here you should emphasize novelty and originality of the present study once again. You compare your results with the ones obtained by other authors, but you need to highlight what you bring in novelty compared to these. Another observation is related to the fact that you cannot include conclusions in this section. Please revise accordingly.
R2: we included our findings in the discussion section, and copied some of the novel findings here:
- Mutants that express (partial) photomorphogenesis characteristics in the dark e.g. cns, pifQ, and spa1/2/3 triple (Figure 5) and mutants expressing a constitutive active PHYB [5], show a severely reduced capacity of AR formation.
300 However, in our experiments, the blue-light effect was entirely dependent on the blue light receptors CRY1 and CRY2, and partially on PHOT1/PHOT2, indicating that pho-tosynthesis played only a minor role.
- Next to auxin biosynthesis, auxin receptivity also determines AR formation. This follows from the reduction in AR in csn mutants (Figure 4).
- A hint for the mechanism by which blue light might regulate AR induction comes from the phot mutant analysis (Figure 3).
A model is now included that highlights all the new results in a context of what is already known.
R3C4: Materials and methods: Not all methods have references. Are they completely novel? If not, please offer references.
R4: references have been added.
R3C5: Conclusions: You need to add them because your Discussions and Results sections are quite long and it is necessary to draw some important aspects regarding your findings in the study. Please offer interpretations of the findings in your study, alongside with perspectives.
R5: A conclusion is added.
All these suggested changes should be performed in order to bring further improvements to the manuscript.
Round 2
Reviewer 3 Report
Dear Authors,
The present study analyzes adventitious rooting under different light conditions in wild type and photomorphogenesis mutants. The authors performed some of the suggested changes after the first round of review. Following specific changes should still be performed:
Major changes
Introduction: I do not see any of the suggested changes in the first round of review. Introduction needs to highlight novelty and originality, it is not enough to find them in the Abstract. They need to be emphasized in this section, as it is not clear for readers. Moreover, in the part you describe the purpose of the study, you should state why the results obtained in this study may prove important. The purpose of the study is still not clear. I also cannot find the exact name of Arabidopsis species that represents the subject of the study. Arabidopsis is a genera and it shouldn’t be used when speaking about a specific species. Rationale for choosing the species and details about it in the context of your study are still not found.
Discussions: I cannot see where suggestions from the previous round were added. I still cannot see novelty and originality emphasized.
All these suggested changes should still be performed in order to bring further improvements to the manuscript.
Author Response
Comments from the Reviewer #3 (R2):
Dear Authors,
The present study analyzes adventitious rooting under different light conditions in wild type and photomorphogenesis mutants. The authors performed some of the suggested changes after the first round of review. Following specific changes should still be performed:
Major changes
C1: Introduction: I do not see any of the suggested changes in the first round of review. Introduction needs to highlight novelty and originality, it is not enough to find them in the Abstract. They need to be emphasized in this section, as it is not clear for readers. Moreover, in the part you describe the purpose of the study, you should state why the results obtained in this study may prove important. The purpose of the study is still not clear. I also cannot find the exact name of Arabidopsis species that represents the subject of the study. Arabidopsis is a genera and it shouldn’t be used when speaking about a specific species. Rationale for choosing the species and details about it in the context of your study are still not found.
R1: - We apologize for misinterpreting this comment. We have revisited the introduction and highlighted what is unknown and thus where novelty from our work arises: line 52-53; 64-68; 80; 82-83; 95-98; 115-130.
- In the last paragraph of the introduction (line 115-117), we explicitly state why the results of this study may prove important.
- We always used Arabidopsis thaliana. We have included this essential piece of information in the title, abstract, and at several positions throughout the text.
- Given the abundance of genetic tools in Arabidopsis thaliana, we have chosen to work with Arabidopsis thaliana, in the ecotype background, in which desired mutants were available (Col-0, Ws or Ler). We have added this reason in line 138.
C2:Discussions: I cannot see where suggestions from the previous round were added. I still cannot see novelty and originality emphasized.
R2: We apologize for this, as we failed to correctly interpret this suggestion in the previous round. We have now attempted to highlight the novelty and originality, by emphasizing our contribution by extra referral to our data in the text. Eg. line 314, 335; 352-354; 375-376; 380
We also provide a clear summary of the data, its meaning and some future perspectives in the conclusions section (line 414-427)
C3: All these suggested changes should still be performed in order to bring further improvements to the manuscript.
R3: We thank the reviewer for being patient, and for helping us to improve our manuscript.
Round 3
Reviewer 3 Report
Dear Authors,
The present study analyzes adventitious rooting under different light conditions in wild type and photomorphogenesis mutants. The authors performed most of the suggested changes after the second round of review. Following specific changes should still be performed:
Minor changes
Introduction: The purpose of the study is still not clear.
Conclusions need to be found at the end of the manuscript.
All these suggested changes should still be performed in order to bring further improvements to the manuscript.
Author Response
Comments from the Reviewer #3 (R3):
Dear Authors,
The present study analyzes adventitious rooting under different light conditions in wild type and photomorphogenesis mutants. The authors performed most of the suggested changes after the second round of review. Following specific changes should still be performed:
Minor changes
C1: Introduction: The purpose of the study is still not clear.
R1: We apologize for not being clear enough. We have now stated explicitly the purpose of the study (line 115-119)
C2: Conclusions need to be found at the end of the manuscript.
R2: Thank you for pointing out that the Conclusions should be after the Materials and Methods. We have moved this section accordingly (line 459-472)
All these suggested changes should still be performed in order to bring further improvements to the manuscript.